# 4-Nitrophenol-Loaded Magnetic Mesoporous Silica Hybrid Materials for Spectrometric Aptasensing of Carcinoembryonic Antigen

**DOI:** 10.3390/mi12101138

**Published:** 2021-09-22

**Authors:** Jin Zhang, Dianping Tang

**Affiliations:** 1Chongqing Vocational Institute of Engineering, Chongqing 402260, China; zhangjin@cqwu.edu.cn; 2Key Laboratory of Analytical Science for Food Safety and Biology, Department of Chemistry, Fuzhou University, Fuzhou 350108, China

**Keywords:** aptasensor, carcinoembryonic antigen, magnetic mesoporous silica, aptamer, controlled release, spectrometric assay, UV–vis absorption spectroscopy, 4-nitrophenol

## Abstract

Aptamer- or antibody-based sensing protocols have been reported for detecting carcinoembryonic antigen (CEA), but most exhibit complicated procedures or multiple reactions. In this work, we developed a one-step aptasensing protocol for the spectrometric determination of CEA based on 4-nitrophenol (4-NP)-loaded magnetic mesoporous silica nanohybrids (MMSNs) for bioresponsive controlled-release applications. To fabricate such a responsive–controlled sensing system, single-stranded complementary oligonucleotides relative to the CEA-specific aptamer were first modified on the aminated MMSN. Thereafter, 4-NP molecules blocked the pores with the assistance of the aptamers via a hybridization reaction. The introduced target CEA specifically reacted with the hybridized aptamer, thus detaching from the MMSN to open the gate. The loaded 4-NP molecules were released from the pores, as determined using ultraviolet–visible (UV–vis) absorption spectroscopy after magnetic separation. Under optimum conditions, the absorbance increased with an increase in the target CEA in the sample and exhibited a good linear relationship within the dynamic range of 0.1–100 ng mL^−1^, with a detection limit of 46 pg mL^−1^. Moreover, this system also displayed high specificity, good reproducibility, and acceptable accuracy for analyzing human serum specimens, in comparison with a commercialized human CEA-enzyme-linked immunosorbent assay (ELISA) kit.

## 1. Introduction

A tumor biomarker, e.g., alpha-fetoprotein (AFP), carcinoembryonic antigen (CEA), cancer antigen 125 (CA 125), or cancer antigen 15-3 (CA 15-3), is anything present in, or produced by, cancer cells in the body, relating to cancer or certain benign (noncancerous) conditions [1,2,3]. CEA produces a cell adhesion molecule consisting of an Ig variable-region-like amino-terminal domain followed by up to six Ig constant-region-like domains, anchored to the cell membrane by either a glycosylphosphatidylinositol moiety or a proteinaceous transmembrane and cytoplasmic domain [4,5]. Under physiologic conditions, only low amounts of soluble CEA can be detected in serum. However, in many different cancers, CEA is highly upregulated [6]. Therefore, the sensitive and specific detection of CEA concentration in human serum would be advantageous for evaluating therapy and detecting recurrence or metastasis.

Recently, protocols based on aptasensors and immunoassays have been developed for the quantitative monitoring of CEA, e.g., radioimmunoassay, enzyme-linked immunosorbent assay (ELISA), fluoroimmunoassay, chemiluminescent immunoassay, and electrochemical immunoassay techniques [7,8]. Yu et al. developed a pressure-based biosensor integrated with a flexible pressure sensor and an electrochromic device for a visual immunoassay of CEA [9]. Tang et al. designed a surface-enhanced Raman-scattering-based lateral flow immunoassay mediated by a hydrophilic–hydrophobic silver (Ag)-modified poly(methyl methacrylate) (PMMA) substrate [10]. Xia et al. utilized a microfluidic magnetic analyte delivery technique for the separation, enrichment, and fluorescence detection of ultratrace quantities of CEA [11]. Gu et al. constructed a sandwich-type electrochemical immunosensor for CEA detection [12]. Ahmed et al. fabricated an electrochemical sensor for the label-free detection of carcinoembryonic antigen based on gold nanoparticles, carbon nano-onions, and single-walled carbon nanotubes [13]. Nakhjavani et al. also developed gold and silver bio/nanohybrid-based electrochemical immunosensors for the ultrasensitive detection of carcinoembryonic antigen [14]. Wu et al. proposed a colorimetric immunoassay based on a coordination polymer composite for the detection of carcinoembryonic antigen [15]. Typically, immunological methods have become the predominant analytical techniques for the quantitative detection of tumor markers, because of the highly specific molecular recognition of the antibody and epitopes of an antigen [16]. In these systems, the assays are usually carried out using the sandwiched immunoreaction mode [17]. The immobilized antibody first captures the corresponding analyte (antigen), and the captured antigen is then detected by another labeled antibody [18]. Despite the sensitivity of conventional immunoassays, they have some limitations, such as multiple labels, a complicated washing procedure, and expensive immunological reagents [19,20,21]. To this end, our motivation in this study is to design a method for the simple and sensitive detection of disease-related biomarkers.

In contrast, aptamers are synthetic oligonucleotides selected from a random nucleotide library using the Systematic Evolution of Ligands by Exponential enrichment (SELEX) method [22]. They show extraordinary analyte binding efficiency that is comparable to that of antigen–antibody reactions [23,24]. In addition, most aptamers are more stable towards oxidation and hydrolysis than antibodies and can be easily modified by the manipulation of terminal functional groups [25]. Li et al. developed a zeolitic imidazolate framework-8 (ZIF-8)-assisted NaYF4:Yb,Tm≅ZnO converter with an exonuclease III-powered DNA walker for a near-infrared light-responsive aptasensor for CEA [26]. Inspiringly, bionanotechnology can provide a new horizon for the development of analytical methods [27]. Qiu et al. designed a bioresponsive release system for the visual fluorescence detection of carcinoembryonic antigen from mesoporous silica nanocontainers mediated by optical color on quantum dot enzyme-impregnated paper [28]. In this regard, mesoporous silica nanoparticles can be utilized as nanocontainers for the loading of indicators or tracers using the corresponding aptamers as the gate. 4-Nitrophenol (4-NP, used as a tracer) can exhibit good spectroscopic characteristics during measurement.

Herein, we report the proof of concept for a simple and sensitive spectrometric aptasensing method for the detection of CEA using 4-nitrophenol-loaded magnetic mesoporous silica hybrid nanomaterials. The tracer, 4-nitrophenol, is initially gated in the pores of magnetic mesoporous silica nanohybrids using CEA-specific aptamers, after being hybridized with the conjugated complementary single-stranded DNA on the surface of mesoporous silica. Upon reaction of the target CEA with the aptamer, the CEA–aptamer complex formed is dissociated from the magnetic mesoporous silica, thus resulting in the opening of the gate. In this case, the loaded 4-nitrophenol molecules are diffused from the pores, which can be determined using UV–vis absorption spectroscopy. The main objective of this work is to explore a simple spectrometric aptasensing protocol for the detection of CEA via coupling with a target-induced controlled-release system.

## 2. Materials and Methods

### 2.1. Chemicals and Reagents

Recombinant human carcinoembryonic antigen (CEA) protein (expression system: wheat germ; tags: GST tag N-terminus; suitable for: ELISA; cat# no.: ab158095; 10 μg) and a recombinant human CEA-enzyme-linked immunosorbent assay (ELISA) kit including CEA standards with different concentrations (sensitivity: 0.2 ng mL^−1^; range: 0.343–250 ng mL^−1^; sample type: cell culture supernatant plasma, serum; detection method: colorimetric; assay type: sandwich, quantitative; reacts with: human; cat# no.: ab99992) were purchased from Abcam (Shanghai, China). All the oligonucleotides, including the CEA-specific aptamer, were synthesized by Sangon Biotech Co., Ltd. (Shanghai, China), and the corresponding sequences were listed as follows: CEA-specific aptamer: 5′-ATA CCA GCT TAT TCA ATT-3′; complementary DNA (cDNA): 5′-HOOC-AAT TGA ATC GTG GTA T-3′ (note that the CEA-specific aptamer was partially complementary with the cDNA for the formation of the arched structures to gate the pores of magnetic mesoporous silica and all the oligonucleotides were first heated to 95 °C for 5 min and then cooled to room temperature prior to use). Magnetic beads consisting of an aqueous dispersion of magnetic iron oxides with a diameter of 100 nm (formulation: suspension in ddH_2_O, autoclaved; concentration: 25 mg mL^−1^) were obtained from Chemicell GmbH (Berlin, Germany). Cetyltrimethylammonium bromide (CTAB), (3-aminopropyl)-triethoxysilane (APTES), 1-ethyl-2-(3-dimethylaminopropyl) carbodiimide hydrochloride (EDC), *N*-hydroxysulfosuccinimide (NHS), and tetraethyl orthosilicate (TEOS) were obtained from Sigma-Aldrich (Merck KGaA, Darmstadt, Germany). All other chemicals were of analytical grade (A.R.) and were used without further purification. Ultrapure water, used in all runs, was prepared using a Millipore water purification system (18.2 MΩ cm). All buffers, including phosphate-buffered saline (PBS) solution, were Sigma-Aldrich products.

### 2.2. Synthesis of Magnetic Mesoporous Silica Nanocontainers (MMSNs)

Aminated magnetic mesoporous silica nanocontainers (MMSNs) were synthesized according to our previous reports [29,30] with minor modifications. Initially, 25 mg of CTAB was added to 5.0 mL of H_2_O/ethanol (1:1, *v*/*v*) containing 2.0 mL of magnetic beads (25 mg mL^−1^) under sonication, to achieve an apparently uniform suspension (~15 min). Thereafter, TEOS (200 μL, undiluted concentration) and APTES (500 μL, undiluted concentration) were injected into the resultant suspension. The mixture was sonicated for another 15 min to form a stable suspension. Following that, NH_4_OH (2.0 mL, original concentration) was added to the suspension under sonication for 4 h at room temperature. Next, 500 μL of TEOS/APTES/ethanol (4:1:40, *v*/*v*/*v*) solution was added to the mixture under the same conditions as for the synthesis of aminated magnetic silica nanospheres. After sonication for another 2 h, the resulting suspension was magnetically separated using an external magnet and washed with ultrapure water/ethanol (3:1, *v*/*v*) three times (note that during this process, only pure silica nanospheres could be removed). The obtained precipitates were redispersed in 5.0 mL of ethanol and refluxed at 90 °C to remove CTAB (note that the refluxing process was repeated five times). Finally, the aminated MMSNs were dried at 60 °C for subsequent use.

### 2.3. Conjugation of Aminated MMSNs with Carboxylated cDNA (cDNA-MMSN)

Complementary DNA-conjugated magnetic mesoporous silica nanoparticles were prepared via typical carbodiimide coupling [31,32]. Initially, 200 μL of 120 μM cDNA was added to 500 μL of distilled water containing 10 mg mL^−1^ of EDC and 10 mg mL^−1^ of NHS, followed by continuous stirring for 60 min at room temperature in order to activate the carboxyl group on the cDNA. Then, 200 μL of 25 mg mL^−1^ of aminated MMSN dispersed into distilled water was added to the mixture under continuous stirring at 150 rpm and left at room temperature for 20 h (note that the amount of carboxylated cDNA should be in excess). After completion of the reaction, the resulting suspension was magnetically separated and washed with PBS (10 mM, pH 7.4) several times. During this process, the activated cDNA was covalently conjugated with the aminated MMSN. Finally, the obtained cDNA-MMSN was dispersed in PBS (10 mM, pH 7.4) for subsequent use.

### 2.4. Loading of 4-Nitrophenol with cDNA-MMSN via the CEA-Specific Aptamer

4-Nitrophenol tracers were loaded into the pores of magnetic mesoporous silica nanoparticles on the basis of a hybridization reaction between the conjugated cDNA and the CEA-specific aptamer. Initially, 20 mg of cDNA-MMSN was dispersed into 2.0 mL of PBS (10 mM, pH 7.4) containing 250 μM of aptamer and 2.0 M 4-nitrophenol (excess). Thereafter, the mixture was gently shaken on a shaker overnight at room temperature. During this process, the CEA-specific aptamer hybridized with the conjugated cDNA on the MMSN to form an omega (Ω) DNA structure for the gating of the pore. Meanwhile, numerous 4-nitrophenol molecules were gated into the pores. Following that, the resultant suspension was separated using an external magnet. The collected bionanoparticles (i.e., aptamer/cDNA-MMSN loaded with 4-nitrophenol) were dispersed into PBS (10 mM, pH 7.4) at a fixed concentration of ~5.0 mg mL^−1^ and stored at 4 °C when not in use. 

### 2.5. Spectrometric Measurement of Aptasensing Platform with Respect to the Target CEA

Figure 1 shows a schematic illustration of 4-nitrophenol-loaded magnetic mesoporous silica hybrid nanomaterials for spectrometric aptasensing of CEA. The assay was carried out as follows: (i) 50 μL of CEA standard/sample at a certain concentration was added to a 100 μL aptamer/cDNA-MMSN suspension loaded with 4-nitrophenol (5.0 mg mL^−1^) and reacted for 50 min at 37 °C in a PCR tube with slight shaking on a shaker; (ii) after magnetic separation, the obtained supernatant fluid was determined using a plate reader (Tecan, Infinite M1000 Pro, Männedorf, Switzerland). The absorption spectra were collected in the range of 250–500 nm, and the absorbance at 318 nm was registered as the spectrometric aptasensing signal relating to the CEA concentration. All determinations were made at least in duplicate. The sigmoidal curves were calculated by mathematically fitting experimental points using Rodbard’s four-parameter function with Origin 10.0 software. Graphs were plotted in the form of absorbance against the logarithm of the CEA concentration. All measurements were carried out at room temperature (25 ± 1.0 °C).

### 2.6. Statistical Analysis

Statistical data analysis was performed using SAS ver. 9.0 and SPSS ver. 9.0 software. Comparisons between dependent variables were determined using analysis of variance (ANOVA), Duncan’s multiple range test, correlation analysis, and multiple regression analysis. Results are expressed as the mean value ± standard deviation (SD) of three determinations, and statistical significance was defined at *p* ≤ 0.05.

## 3. Results and Discussion

### 3.1. Design of Spectrometric Aptasensing Platform

In this system, magnetic mesoporous silica nanoparticles are used as nanocontainers for the loading of the 4-nitrophenol tracer. One merit of using magnetic beads is that they enable the rapid separation and purification of bionanocomposites after synthesis, with the assistance of an external magnet. Silica nanoshells are formed on the surface of the magnetic beads via an in situ coating approach using the wet-chemistry method during the synthesis. The formation of mesoporous silica is based on the doped CTAB. 4-Nitrophenol, with an absorbance maximum at 318 nm, is used for display devices, solar cells, electrolytic capacitors, and so on, and is employed as a tracer for the spectrometric measurements. Carboxylated complementary single-stranded DNA (cDNA) is covalently conjugated with the surface of the aminated MMSNs via a typical carbodiimide coupling method, with the help of EDC and NHS. The tracer, 4-nitrophenol, is loaded into the pores on the basis of the formed omega double-stranded DNA structures after reaction of cDNA with the CEA-specific aptamer. In the absence of the target CEA, 4-nitrophenol molecules are gated in the pores; thus, no absorbance is observed in the supernatant fluids. Upon addition of the target CEA, the hybridized aptamer reacts specifically with the analyte and detaches from the MMSN, thereby opening the pores to release the loaded 4-nitrophenol molecules into the solution. In this case, the collected supernatant containing the released 4-nitrophenol can be determined by UV–vis absorption spectrometry. The absorbance depends on the CEA concentration in the sample. By evaluating the change in the absorbance, we can calculate the CEA level.

### 3.2. Characterization of Magnetic Mesoporous Silica Nanostructures

As described above, the aptasensing platform was fabricated using the as-synthesized magnetic mesoporous silica nanocontainers. Figure 1A shows a typical transmission electron microscopic (TEM; Mdel H-7650, Hitachi Instruments, Tokyo, Japan) image of the as-purchased magnetic beads with an average size of 100 nm in diameter. In contrast, aminated magnetic mesoporous silica nanoscales exhibited good dispersion in aqueous solution (Figure 1B). Moreover, numerous pores could be observed from the high-resolution TEM image (Figure 1B, inset). Furthermore, the nitrogen adsorption–desorption isotherm gave a type IV isotherm curve with a relatively large surface area (Figure 1C) and a pore size of 5.1 nm (Figure 1C, inset). These large pores can be used for loading numerous 4-nitrophenol molecules. In addition, magnetic measurements were made using a vibrating sample magnetometer (VSM). The hysteresis loops for the magnetic beads before and after functionalization with silica nanostructures are plotted in Figure 1D. Clearly, the magnetization of magnetic mesoporous silica was less than that of the magnetic beads alone (curve ‘b’ vs. curve ‘a’), suggesting that the silica shell formed a coating on the magnetic beads and the magnetization could still remain after modification.

### 3.3. Chacteristics of 4-NP-Loaded MMSN with cDNA and Aptamer

In order to develop a spectrometric aptasensing system, the 4-NP-loaded MMSN with cDNA and aptamer must be successfully synthesized during preparation. First, we used microelectrophoresis (Microelectrophoresis Apparatus Mk II, Rank Brothers Ltd., Cambridge, UK) to investigate the zeta potentials of nanocontainers after each step (Figure 2). The as-purchased magnetic beads displayed a negative surface potential (column ‘a’) since silica nanoparticles have a high proportion of surface silanol groups in aqueous solution. In contrast, the aminated magnetic mesoporous silica showed a positive surface potential (column ‘b’) due to the presence of amino groups. When the carboxylated cDNA molecules were covalently conjugated with the aminated MMSNs, the surface potential became negative (column ‘c’), which mainly derived from negatively charged oligonucleotides. Moreover, the surface potential became more negative when CEA-specific aptamers hybridized with cDNA-functionalized MMSNs (column ‘d’). These results give a preliminary indication that cDNA and aptamers can be conjugated with the carboxylated magnetic mesoporous silica nanocontainers.

### 3.4. Feasibility Evaluation of Spectrometric Aptasensing Platform

To monitor the feasibility of the aptasensing platform, the 4-NP-encapsulated aptamer/cDNA/MMSNs were used for the detection of the target CEA (1.0 ng mL^−1^ used as an example). The supernatant obtained after magnetic separation in every experiment was determined using UV–vis absorption spectrometry (Figure 3). Curve ‘a’ represents the UV–vis spectrum of pure 4-nitrophenol alone. In contrast, only one characteristic peak was observed at ~260 nm for the 4-NP-encapsulated aptamer/cDNA/MMSNs (curve ‘b’). This peak originated from the conjugated c-DNA and aptamer, whereas no characteristic absorption peak at 318 nm for 4-nitrophenol appeared. These results further reveal that 4-nitrophenol molecules could be firmly gated in the pores using the formed omega double-stranded DNA between the cDNA and the aptamer. Upon addition of 1.0 ng mL^−1^ of CEA to PBS (10 mM, pH 7.4) containing 4-NP-encapsulated aptamer/cDNA/MMSNs, significantly, an obvious characteristic absorption peak at 318 nm was acquired in the supernatant (curve ‘c’) which stemmed from the 4-nitrophenol molecules released from the pores. Therefore, the as-prepared 4-NP-encapsulated aptamer/cDNA/MMSNs could be applied, in a preliminary sense, for the detection of the target CEA on the basis of a target-induced controlled-release system.

### 3.5. Optimization of Experimental Conditions

In order to firmly gate the 4-nitrophenol molecules in the pores, the concentration of CEA-specific aptamer is crucial for the formation of omega double-stranded DNA structures. In this case, the as-prepared 4-NP-encapsulated aptamer/cDNA/MMSNs, using aptamers of varying concentration, were dispersed in PBS (10 mM, pH 7.4) and stored at room temperature for 60 min with slight shaking on a shaker. Following this, the supernatants were collected after magnetic separation and characterized using UV–vis absorption spectroscopy. As shown in Figure 4A, the maximum absorbance at 318 nm decreased with increasing aptamer concentration and almost tended to zero above a concentration of 200 μM. To ensure adequate conjugation, we selected the 250 μM CEA-specific aptamer for the reaction with the labeled cDNA.

Another factor influencing the analytical performance of the aptasensing system is the reaction time for the target CEA with 4-NP-encapsulated aptamer/cDNA/MMSNs. A short reaction time is not favorable for the release of 4-nitrophenol. As seen in Figure 4B, the absorbances increase with increasing reaction time toward 1.0 ng mL^−1^ of CEA, tending to level off after 50 min. To save time in the assay, 50 min was chosen as the reaction time for the detection of the target CEA.

### 3.6. Spectroscopic Response of Aptasensing Platform with Respect to Target CEA Standards

Under optimum conditions, 4-NP-encapsulated aptamer/cDNA/MMSNs were used to determine CEA standards with different concentrations via coupling with a controlled-release system. Figure 5A shows UV–vis absorption spectroscopic diagrams for the developed aptasensor with CEA standards of varying concentrations. The absorbances increased with increasing CEA concentration in the samples. However, the changes in the absorbance were not obvious within the low (from 0.01 ng mL^−1^ to 0.1 ng mL^−1^) or high (from 100 ng mL^−1^ to 1000 ng mL^−1^) concentration ranges. Inspiringly, a good linear relationship between absorbance and the decimal logarithm of CEA concentration was seen within the dynamic range from 0.1 ng mL^−1^ to 100 ng mL^−1^ (Figure 5B): the regression equation was y (a.u.) = 0.24 × logC_[CEA]_ + 0.31 (ng mL^−1^, r = 0.9931, n = 7). The limit of detection (LOD) was estimated as 46 pg mL^−1^ at a signal-to-noise ratio of 3S_B_ (where S_B_ is the standard deviation of the blank, n = 11). Since the threshold of CEA in human serum is 3.0 ng mL^−1^, our aptasensing strategy can meet the requirements for clinical diagnosis.

### 3.7. Reproducibility, Specificity and Stability

The reproducibility and precision characteristics of the spectrometric aptasensor were evaluated by applying the aptasensor to the detection of three CEA standards with different concentrations, on the basis of same-batch or various-batch 4-NP-aptamer/cDNA/MMSNs. The judgement was based on the relative standard deviation (RSD) during the five measurements. Table 1 represents the experimental results, using 0.5, 1.0 and 10 ng mL^−1^ of CEA as examples. Clearly, all RSDs were less than 15% (n = 5), indicating good reproducibility and precision.

The specificity of the spectrometric aptasensor was studied using our method to monitor other possible biomarkers, e.g., alpha-fetoprotein (AFP), neuron-specific enolase (NSE), human chorionic gonadotropin (hCG), cancer antigen 125 (CA 125), and CA 15-3. In this case, the absorbance at a low concentration of the target CEA (0.1 ng mL^−1^) was compared with the absorbance at a high concentration of non-target biomarkers (100 ng mL^−1^ or U mL^−1^). As indicated in Figure 6, the absorbances for non-target biomarkers, including AFP, NSE, hCG, CA 125, and CA 15-3, were close to zero. In contrast, strong absorbances could be observed in the presence of the target CEA. In addition, a mixture of non-target biomarkers with the target CEA did not cause a significant change in the absorbance relative to the analyte alone. These results reveal that our system has good specificity and selectivity towards the target CEA.

In addition, the stability of the prepared 4-NP-aptamer/cDNA/MMSNs was studied under storage at 4 °C. The evaluation was carried out by assaying the absorbance for 1.0 ng mL^−1^ of CEA every two months. Experimental results indicated that the absorbance persevered at 98.9%, 97.4%, 97.1%, 95.9%, 94.8%, and 93.7% of the initial signal, indicating good storage stability. 

### 3.8. Analysis of Real Samples and Interlaboratory Validation

In this study, we collected 15 real human serum samples containing the target CEA from Fujian Tumor Hospital (Fuzhou, China). Referring to the rules of the local ethics committee, informed consent was obtained from all patients who provided specimens. These samples were initially determined using our developed spectrometric aptasensor and a commercial human CEA ELISA kit (for reference). The obtained results are shown in Figure 7. Comparison of the experimental results obtained with these two methods was performed via a least-squares regression method. The regression line was fitted to y = (1.0014 ± 0.0142)x + (−0.2797 ± 0.6209) (R^2^ = 0.9986, n = 15), where x stands for CEA concentrations estimated with the spectrometric aptasensor and y stands for those estimated with the reference procedure. The standard deviations of the slope and intercept were given in the regression equation. The correlation between the two methods was monitored using t-tests for comparison of the experimental values of the intercept and slope to the ideal situation of zero intercept and a slope of 1. Therefore, no significant differences were encountered at the 0.05 significance level, revealing a good agreement between the two methods.

## 4. Conclusions

In conclusion, a simple spectrometric aptasensing method was successfully designed to monitor the target CEA via coupling with a target-induced controlled-release system. The as-synthesized magnetic mesoporous silica nanostructures were used for loading 4-nitrophenol tracers. The designed spectrometric aptasensing method could exhibit high sensitivity, good reproducibility, and high selectivity. In addition, acceptable analytical results for the determination of real samples were achieved using a commercial human CEA ELISA kit as the reference. Although our system was applied to the target CEA, it is easily extended to determine the presence other proteins by controlling the conjugated single-stranded aptamer.

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
