# Peer review of "4-Nitrophenol-Loaded Magnetic Mesoporous Silica Hybrid Materials for Spectrometric Aptasensing of Carcinoembryonic Antigen"

_micromachines, 2021, doi:10.3390/mi12101138_

Round 1

Reviewer 1 Report

The manuscript presents a spectrometric aptasensing system. In general, the aim is clearly stated and appropriate experimental methods have been used to characterize the samples, but there are some gaps in the text that need to be corrected and clarified before it can be published.

  • I think that the term used by the authors "new spectrometric aptasensing method" in the conclusion is inappropriate. The authors use a well-known method of monitoring the dependence of the intensity of a certain characteristic absorption band of the analyte on its concentration. The novelty they can claim is aptly mentioned in the title – authors propose “Materials for Spectrometric Aptasensing of Carcinoembryonic  ”
  • The instruments used to measure the absorption spectra must be specified. In part 2.5. The absorbance at ~ 318 nm was collected and registered as the spectrometric aptasensing signal relative to CEA concentration. All determinations were made at least in duplicate.", while the presented absorption spectra are in the range of 250-500 nm.
  • On page 5 it is mentioned that the characteristic absorption band is ~ 320 nm, while in the text related with UV-Vis absorption is discussed the band position at 318 nm. This difference is insignificant, but it is better to harmonize the data in the text in terms of better presentation.
  • The authors need to specify how the sample concentration is controlled. What is the error in determining it?
  • It should be specified what is the magnification of the image presented in the inset of fig. 1B. Accordingly, in the text on page 6 you can specify the interval of the pore size.
  • The legend in Figure 5A is inappropriate and needs to be improved.

Author Response

Reviewer: 1

The manuscript presents a spectrometric aptasensing system. In general, the aim is clearly stated and appropriate experimental methods have been used to characterize the samples, but there are some gaps in the text that need to be corrected and clarified before it can be published.

  • I think that the term used by the authors "new spectrometric aptasensing method" in the conclusion is inappropriate. The authors use a well-known method of monitoring the dependence of the intensity of a certain characteristic absorption band of the analyte on its concentration. The novelty they can claim is aptly mentioned in the title – authors propose “Materials for Spectrometric Aptasensing of Carcinoembryonic ”

Re: Many thanks for your excellent suggestion! They were corrected throughout this text.

  • The instruments used to measure the absorption spectra must be specified. In part 2.5. The absorbance at ~ 318 nm was collected and registered as the spectrometric aptasensing signal relative to CEA concentration. All determinations were made at least in duplicate.", while the presented absorption spectra are in the range of 250-500 nm.

Re: The instruments used to measure the absorption spectra was “Tecan, Infinite M1000 Pro, Switzerland”, and it was added in the revision text [p.4, 2.5 section]. Meanwhile, this sentence was corrected to “The absorption spectra were collected in the range of 250 – 500 nm, and the absorbance at 318 nm was registered as the spectrometric aptasensing signal relative to CEA concentration” in 2.5 section.

  • On page 5 it is mentioned that the characteristic absorption band is ~ 320 nm, while in the text related with UV-Vis absorption is discussed the band position at 318 nm. This difference is insignificant, but it is better to harmonize the data in the text in terms of better presentation.

Re: Sorry for our mistake! They were corrected to “318 nm” throughout the text. Thanks again!

  • The authors need to specify how the sample concentration is controlled. What is the error in determining it?

Re: Target CEA standards with different concentrations were acquired from the as-purchased human CEA ELISA kit, which was described in 2.1 section [p.3, 2.1 section]. Each data point represents the average value obtained from three measurments; The error bars represent the 95% confidence interval of the mean for y-axis concentrations. They were described in the legends of Figures.  

  • It should be specified what is the magnification of the image presented in the inset of Fig. 1B. Accordingly, in the text on page 6 you can specify the interval of the pore size.

Re: The magnification of the image presented in the inset of Figure 1B was 1.5 fold, and it was added in the legend of Figure 1B [p.4, Figure 1]. The interval of pore size was 5.1 nm, which was described in the text on page 6.

  • The legend in Figure 5A is inappropriate and needs to be improved.

Re: Thanks! It was corrected to “UV-vis absorption spectra of the obtained supernatants after the aptasensing platform reacted with different-concentration CEA standards” in the text [p.8, legend of Figure 5A].

Reviewer 2 Report

The authors Jin Zhang et al in here present a one-step aptasensing protocol for spectrometric determination of CEA based on 4-nitrophenol (4-NP)-loaded magnetic mesoporous silica nano-hybrids (MMSNs) for bioresponsive controlled-release application. And they exploited methods that the loaded 4-NP molecules were released from the pores, which could be determined by using UV-vis absorption spectroscopy after magnetic separation. Under optimum conditions, the absorbance increased with the increasing of target CEA in the sample, and exhibited a good linear relationship within the dynamic range of 0.1 – 100 ng mL-1 with a detection limit of 46 pg mL-1. Moreover, this system also displayed high specificity, good reproducibility and acceptable accuracy for analyzing human serum specimens in comparison with commercialized human CEA ELISA kit. Recently, there is an incremental interest for tools capable of rapid and reliable detection of carcinoembryonic antigen. This topic may attract many readers. However, the manuscript with poor writing is poorly organized. It’s highly recommended to be polished and revised by native speakers before it’s published.

  1. Compared to recently published papers such as Gu, Xuefang, et al. "Electrochemical detection of carcinoembryonic antigen." Biosensors and Bioelectronics102 (2018): 610-616.; Rizwan, Mohammad, et al. "AuNPs/CNOs/SWCNTs/chitosan-nanocomposite modified electrochemical sensor for the label-free detection of carcinoembryonic antigen." Biosensors and Bioelectronics107 (2018): 211-217.; Wu, Sixuan, et al. "A colorimetric immunoassay based on coordination polymer composite for the detection of carcinoembryonic antigen." ACS applied materials & interfaces 11.46 (2019): 43031-43038.; Nakhjavani, Sattar Akbari, et al. "Gold and silver bio/nano-hybrids-based electrochemical immunosensor for ultrasensitive detection of carcinoembryonic antigen." Biosensors and Bioelectronics 141 (2019): 111439., the principle and innovation of this paper should be explained more in details. There are many related topic and researches, the innovation of this paper is highly recommended to be described in more details.
  2. There are many abbreviations in use. Please show the full names before use them.
  3. There are some grammar errors which should be revised carefully.
  4. The part of introduction is recommended to be re-organized.
  5. There are two stars showing after authors, but only listing one correspondence. Please carefully revise it and avoid any possible necessary correction after its publication.

Author Response

Reviewer: 2

The authors Jin Zhang et al in here present a one-step aptasensing protocol for spectrometric determination of CEA based on 4-nitrophenol (4-NP)-loaded magnetic mesoporous silica nano-hybrids (MMSNs) for bioresponsive controlled-release application. And they exploited methods that the loaded 4-NP molecules were released from the pores, which could be determined by using UV-vis absorption spectroscopy after magnetic separation. Under optimum conditions, the absorbance increased with the increasing of target CEA in the sample, and exhibited a good linear relationship within the dynamic range of 0.1 – 100 ng mL-1 with a detection limit of 46 pg mL-1. Moreover, this system also displayed high specificity, good reproducibility and acceptable accuracy for analyzing human serum specimens in comparison with commercialized human CEA ELISA kit. Recently, there is an incremental interest for tools capable of rapid and reliable detection of carcinoembryonic antigen. This topic may attract many readers. However, the manuscript with poor writing is poorly organized. It’s highly recommended to be polished and revised by native speakers before it’s published.

  1. Compared to recently published papers such as Gu, Xuefang, et al. "Electrochemical detection of carcinoembryonic antigen." Biosensors and Bioelectronics102 (2018): 610-616.; Rizwan, Mohammad, et al. "AuNPs/CNOs/SWCNTs/chitosan-nanocomposite modified electrochemical sensor for the label-free detection of carcinoembryonic antigen." Biosensors and Bioelectronics107 (2018): 211-217.; Wu, Sixuan, et al. "A colorimetric immunoassay based on coordination polymer composite for the detection of carcinoembryonic antigen." ACS applied materials & interfaces46 (2019): 43031-43038.; Nakhjavani, Sattar Akbari, et al. "Gold and silver bio/nano-hybrids-based electrochemical immunosensor for ultrasensitive detection of carcinoembryonic antigen." Biosensors and Bioelectronics141 (2019): 111439., the principle and innovation of this paper should be explained more in details. There are many related topic and researches, the innovation of this paper is highly recommended to be described in more details.

Re: Many thanks for your excellent suggestion! Typically, immunological methods have become the predominant analytical techniques for quantitative detection of tumor markers because of the highly specific molecular recognition of antibody and epitopes of an antigen. In these systems, the assays are usually carried out by using the sandwiched immunoreaction mode. The immobilized antibody first captures the corresponding analyte (antigen), and the captured antigen is then detected by another labeled antibody. Despite the sensitivity of conventional immunoassays, they have some limitations, such as multiple labels, complicated washing procedure and expensive immune reagents. To this end, our motivation of this study is to design a method for simple and sensitive detection of disease-related biomarkers. They including these literatures were added and described in the introduction [p.2, first paragraph].

  1. There are many abbreviations in use. Please show the full names before use them.

Re: All abbreviations were defined at the first usage throughout the text.

  1. There are some grammar errors which should be revised carefully.

Re: We have carefully checked and modified this manuscript.

  1. The part of introduction is recommended to be re-organized.

Re: Thanks a lot! Partial irrelative information was removed in the revision text.

  1. There are two stars showing after authors, but only listing one correspondence. Please carefully revise it and avoid any possible necessary correction after its publication.

Re: Corrected in the revision text.

Round 2

Reviewer 1 Report

I reviewed the revised version of the manuscript and the authors' responses. Thanks to the authors for their comprehensive answers. I found that after the clarifications made by the authors, the manuscript has all the necessary qualities to be published without further corrections.

Reviewer 2 Report

The authors Jin Zhang et al present interesting data about 4-Nitrophenol-Loaded Magnetic Mesoporous Silica Hybrid Materials for Spectrometric Aptasensing of Carcinoembryonic Antigen. The authors have carefully reversed their manuscripts and answer the reviewer’s questions. The manuscript is well written, has important scientific value, and should be of great interest to the readers. And the results are well presented and the statistical analysis would help a lot for related readers. However, Micromachines ranks at JCR - Q2 (Instruments & Instrumentation) and CiteScore - Q2 (Mechanical Engineering) with high IF at 2.891 now. The novelty of the manuscript may not meet the currently high-quality requirement of the journal. If editor strongly recommend to publish it, I have no objection to doing this.